# Bayesian model selection for multilevel models using integrated likelihoods

**Tom Edinburgh**[1]*, **Ari Ercole**[2☯], **Stephen Eglen**[1☯]

**1** Department of Applied Mathematics and Theoretical Physics, University of Cambridge, Cambridge, United Kingdom, **2** Cambridge Centre for Artificial Intelligence in Medicine and Division of Anaesthesia, Department of Medicine, University of Cambridge, Cambridge, United Kingdom

☯ These authors contributed equally to this work.
* te269@cam.ac.uk

## Abstract

Multilevel linear models allow flexible statistical modelling of complex data with different levels of stratification. Identifying the most appropriate model from the large set of possible candidates is a challenging problem. In the Bayesian setting, the standard approach is a comparison of models using the model evidence or the Bayes factor. Explicit expressions for these quantities are available for the simplest linear models with unrealistic priors, but in most cases, direct computation is impossible. In practice, Markov Chain Monte Carlo approaches are widely used, such as sequential Monte Carlo, but it is not always clear how well such techniques perform. We present a method for estimation of the log model evidence, by an intermediate marginalisation over non-variance parameters. This reduces the dimensionality of any Monte Carlo sampling algorithm, which in turn yields more consistent estimates. The aim of this paper is to show how this framework fits together and works in practice, particularly on data with hierarchical structure. We illustrate this method on simulated multilevel data and on a popular dataset containing levels of radon in homes in the US state of Minnesota.

## Introduction

Multilevel models provide a generalisation of linear models to settings in which the model parameters (e.g. regression coefficients) are in some way stratified by groups within the population [1]. For example, individuals in the population may belong to a much smaller set of groups or clusters, and data may be available on the level of the individual and the level of the group. Such hierarchical data structures occur naturally in a wide array of scientific applications, examples of which include phylogenetics, education, healthcare and medicine [2, 3]. This concept can be arbitrarily extended to any number of groupings that exist within the population, either hierarchically or without nesting. A simple linear model that does not include the multilevel structure is generally regarded as an inferior model choice in such situations as it neglects information inherently within the group structure. Instead, multilevel models explicitly model at each level of granularity. A wide variety of structures are possible, which raises an important question: how may we identify an optimal model structure from a number

**Data Availability Statement:** All data and source files are available from the accompanying repository at Zenodo (https://doi.org/10.5281/zenodo.7314381). The working version of the repository is available at GitHub (https://github.

com/tedinburgh/model-evidence-with-integrated-likelihood).

**Funding:** TE is funded by Engineering and Physical Sciences Research Council (EPSRC) National Productivity Investment Fund (NPIF) EP/S515334/1, reference 2089662. The funders had no role in study design, data collection and analysis, decision to publish, or preparation of the manuscript. https://gow.epsrc.ukri.org/NGBOViewGrant.aspx?GrantRef=EP/S515334/1.

**Competing interests:** The authors have declared that no competing interests exist.

**Abbreviations:** ABC, Approximate Bayesian computation; AIC, Akaike information criterion; MCMC, Markov Chain Monte Carlo; SMC, Sequential Monte Carlo.

of competing hypothesised models? For example, should we include hierarchical structure, and should we prefer a multilevel model with varying intercepts or both varying slopes and intercepts? The answer to this question is generally context-specific, relating to the overarching goals of an analysis, e.g. inference or prediction, and to any prior knowledge the researcher has about the problem. In conjunction with this, there exists an array of criteria that can be used to compare the suitability of two separate models. For example, in the frequentist setting, the mostly widely used is the Akaike information criterion (AIC) [4], though other approaches include false-discovery rate [5] and likelihood ratio tests [6, 7].

In this work, we instead focus on Bayesian approaches to model selection, where the usual strategy is to calculate the Bayes factor of two competing models. This is defined as the ratio of the model evidence for each model, where the model evidence is the likelihood integrated over all model parameters with respect to the prior. A key advantage of using the model evidence for model comparison is that it implicitly discourages overfitting by penalising model complexity, since including additional parameters will increase the dimension of the parameter space to be integrated over. By way of contrast, the penalty on model complexity has to be artificially introduced in the AIC framework.

Direct calculation of the model evidence and Bayes factor is well-established for linear models under a normal-inverse-gamma prior (e.g. [8]), but cannot be obtained analytically for multilevel models, as the integral is intractable. As a result, the Bayes factor must be estimated, either by directly approximating the integral as a sum, for instance using importance sampling [9] or sequential Monte Carlo [10], or by jointly estimating posterior probabilities of proposed models through approximate Bayesian computation (ABC) methods, or by numerical methods [11, 12]. In ABC methods, a hierarchical Markov Chain Monte Carlo (MCMC) sampling scheme alternates between two sampling steps, first across the indices denoting each model and then for model parameters of the current chosen model. This requires specification of prior probabilities for the individual models, in addition to priors for the parameters of each model. The relative acceptance frequencies in the chain for the model index then provides an approximation to the posterior probabilities for the models. This, alongside the given priors, allow an estimation of the Bayes factor that bypasses the need to estimate model evidence for each model. A key challenge in such an approach though, is to ensure sufficient mixing in the MCMC chain for the model index, since if the MCMC spends too long exploring only one model, the resulting extreme autocorrelation biases the posterior probability estimates. There are several approaches to this ABC framework, including reversible-jump MCMC [13] and product-space MCMC [14]. In contrast to this imposed hierarchical structure of models, sequential Monte Carlo (SMC) can be run separately on each model, as a by-product of the algorithm is a direct estimate of the log model evidence. This is achieved by a combination of Metropolis-Hastings and importance sampling, in which the likelihood is optimised using a simulated annealing process. Whilst these MCMC approaches are widely used, the estimates tend to suffer dramatically in high-dimensional settings, due to challenges in adequately sampling associated complex high-dimensional parameter spaces.

This motivates the approach to the estimation of Bayes factors that we take here, using partially-integrated likelihoods instead of full likelihoods. We treat (potentially high-dimensional) non-variance parameters, such as the regression coefficients in the model, as nuisance parameters, and we analytically integrate these out with respect to conjugate Gaussian priors, since this reduces the dimension of the problem. This reduces the full likelihood on all parameters to an integrated likelihood on only variance parameters. We can then estimate the model evidence by returning to sequential Monte Carlo (or any of the aforementioned estimation methods), which yields improved results, reduces the bias and variance in estimates, and typically improves computational efficiency.

We illustrate our technique using both simulated data and *Minnesota radon contamination* dataset introduced by [1]. For the former, we simulate four datasets with multilevel structure that correspond to four models described in Methods and then estimate the model evidence for each model and each dataset. In the latter, we estimate the model evidence for various multilevel models proposed by the authors in [1]. This dataset contains measurements of the radon level in houses in the US state Minnesota, as well as predictors at the individual house level and at the county level. The grouping of houses within counties provides an inherent hierarchical structure. As radon is a carcinogen, identifying areas with higher concentrations of radon may be an important consideration in decision-making for homeowners and county authorities. The *Minnesota radon contamination* dataset has been used by several software packages to illustrate multilevel modelling approaches, such as in the Python module PyMC [15, 16].

## Methods

As multilevel linear models are a generalisation of linear models, we can also view a simple linear model as the single-level case within the multilevel framework. For both clarity and computational reasons, we consider linear models and multilevel linear models separately, first summarising notation and then providing the integrated likelihoods in each case, given suitable priors. Multilevel linear models are often interchangeably described as mixed models, where fixed and random effects are equivalent to the population-wide and group-specific variables. We use the vocabulary of multilevel linear models, to mirror the work of [1] as this allows for higher-level generalisation. We provide open-access code for our work at [17]. In this code, we use PyMC for SMC sampling, given the full likelihoods and the integrated likelihoods that we have derived.

### Definitions and notation

We first describe a linear model, in a setting with no multilevel structure. We use $\mathcal{D}$ to denote the data, which contains the observations $(y_i, x_i)$, for $i = 1, \ldots, n$. The independent variables $x_i$ are generally assumed to be vector-valued, with dimension $d$, and we denote the corresponding regression coefficient vector as $\beta$. We assume this contains an intercept term (i.e. the first element of $x_i$ is 1 for all $i$). We focus on the subset of generalised linear models with normal distribution and identity link, which may be considered to be the simplest case for continuous observations, $y_i$. In a Bayesian setting, we require prior distributions for each model parameter, in this case the coefficient $\beta$ and variance parameter $\sigma^2$, to fully define a model. For a fixed-variance multivariate normal distribution likelihood, the conjugate prior for the mean is another multivariate normal distribution. Therefore, we choose to assign a prior of this form for $\beta$. In this section, we will not need specify a distributional form of the prior for $\sigma^2$ (though we later use an inverse-gamma prior). A linear model, denoted $\mathcal{M}$, is:

$$\mathcal{M}: \quad y_i = \beta^T x_i + \epsilon_i, \ \epsilon_i \sim \mathcal{N}(0, \sigma^2)$$
$$\beta \sim \mathcal{N}(\mu, \Sigma), \ \sigma^2 \sim P_{\sigma^2} \tag{1}$$

The parameters of this linear model are denoted by $\theta = (\beta^T, \sigma^2)^T$. In addition to $\theta$, we have various hyperparameters, which include $\mu$, $\Sigma$ and any belonging to the unspecified distribution $P_{\sigma^2}$. Given our choice of multivariate normal prior for $\beta$, we could arbitrarily eliminate the mean $\mu$ by translating the data: $y_{ij} \mapsto y'_{ij} = y_{ij} + \mu^T x_{ij}$ (though we would need to factor this into the interpretation of any results). However, we generally assume the data have been normalised or centred and scaled for computational reasons. MCMC sampling tends to be more

efficient with such preprocessed data, because of reduced auto-correlation in sampling chains, and a translation of the data to eliminate the $\mu$ may conflict with this. We assume independence of priors, so the prior for $\theta$ is the product of the individual priors for $\beta$ and $\sigma^2$. An alternative specification sets a normal-inverse-gamma distribution for the joint prior distribution of $\beta$ and $\sigma^2$, i.e. $\beta, \sigma^2 \sim \mathcal{NIG}(a, b, \mu, \Sigma)$, or $\beta|\sigma^2 \sim \mathcal{N}(\mu, \sigma^2\Sigma)$, $\sigma^2 \sim \mathcal{IG}(a, b)$. Assuming such a relationship between $\beta$ and $\sigma^2$ has computational benefits, as this is conjugate for both parameters, meaning it is possible to fully integrate out over both parameters to get an analytic expression for the model evidence. However, while this prior is convenient, it is generally not useful or realistic in practice [8, 18], and the conjugate nature of the prior does not extend to multilevel models. Given a model $\mathcal{M}$ with parameters $\theta$, we define the following:

- Likelihood, given parameters $\theta$: $p(\mathcal{D}|\mathcal{M}, \theta)$

- Prior distribution function for $\theta$: $p(\theta|\mathcal{M}) = p(\beta|\mathcal{M})p(\sigma^2|\mathcal{M})$

- Integrated likelihood, with integration over $\beta$: $p(\mathcal{D}|\mathcal{M}, \sigma^2) = \int_{\mathcal{R}^d} p(\mathcal{D}|\mathcal{M}, \beta, \sigma^2)p(\beta|\mathcal{M})d\beta$

- Model evidence (or marginal likelihood): $p(\mathcal{D}|\mathcal{M}) = \int_{\Theta} p(\mathcal{D}|\mathcal{M}, \theta)p(\theta|\mathcal{M})d\theta$

- Akaike information criterion: $\mathrm{AIC} = 2k - 2\max_{\theta \in \Theta} \log p(\mathcal{D}|\mathcal{M}, \theta)$, where $k$ is the number of unconstrained parameters.

We now extend this notation to a multilevel linear model. We denote the data, $\mathcal{D}$, as $(y_{ij}, x_{ij}, z_j)$, for the $i^{\text{th}}$ observation in group $j$, with $i = 1, \ldots, n_j$, $j = 1, \ldots, J$ and $\Sigma_j n_j = n$. As before, we focus on the normal-identity case, and we assume variables at the individual-level, $x_{ij}$, and group-level, $z_j$, are vector-valued, with corresponding $d$-dimensional individual-level and $m$-dimensional group-level regression coefficients, $\beta$ and $\alpha$ respectively. The multilevel framework contains a model for each level of the data, as below:

$$\mathcal{M}: \quad \begin{aligned} y_{ij} &= \beta^T x_{ij} + u_j + \epsilon_{ij}, \ \epsilon_{ij} \sim \mathcal{N}(0, \sigma_y^2) \\ u_j &= \alpha^T z_j + \eta_j, \ \eta_j \sim \mathcal{N}(0, \sigma_\eta^2) \\ \beta &\sim \mathcal{N}(\mu_\beta, \Sigma_\beta), \ \alpha \sim \mathcal{N}(\mu_\alpha, \Sigma_\alpha), \ \sigma_y^2 \sim P_{\sigma_y^2}, \ \sigma_\eta^2 \sim P_{\sigma_\eta^2} \end{aligned} \quad (2)$$

We could straightforwardly introduce higher-level groups in analogous manner, though we will not elaborate on this here. It is also worth mentioning that the multilevel linear model can be rewritten as a single-level linear model with correlated errors [19]. For our purposes, it is more convenient to retain the multilevel formulation and, furthermore, to absorb the group-level variables, $z_j$, and group-level regression coefficients, $\alpha$, into their individual-level counterparts, i.e. $(x_{ij}^T, z_j^T) \mapsto x_{ij}^T$ and $(\beta^T, \alpha^T) \mapsto \beta^T$. The prior for the combined regression coefficient has mean $(\mu_\beta^T, \mu_\alpha^T) = \mu^T$ and block diagonal covariance matrix $\mathrm{diag}(\Sigma_\beta, \Sigma_\alpha) = \Sigma$. Instead of the group-level $u_j$, we now model $\eta_j$ as a group-level deviation from the 'population average', and we consider the $J$-dimensional vector $\eta = (\eta_0, \ldots, \eta_J)$ as an additional nuisance variable to integrate out. We then rewrite the above model as:

$$\mathcal{M}: \quad \begin{aligned} y_{ij} &= \beta^T x_{ij} + \eta_j + \epsilon_{ij}, \ \epsilon_{ij} \sim \mathcal{N}(0, \sigma_y^2), \ \eta_j \sim \mathcal{N}(0, \sigma_\eta^2) \\ \beta &\sim \mathcal{N}(\mu, \Sigma), \ \sigma_y^2 \sim P_{\sigma_y^2}, \ \sigma_\eta^2 \sim P_{\sigma_\eta^2} \end{aligned} \quad (3)$$

We now have model parameters $\theta = (\beta^T, \sigma_y^2, \sigma_\eta^2)^T$, and model hyperparameters $\mu, \Sigma$ and those from the unspecified distributions $P_{\sigma_y^2}$ and $P_{\sigma_\eta^2}$. The remaining quantities introduced above are much the same, except:

- Model evidence: $p(\mathcal{D}|\mathcal{M}) = \int_{\Theta \times \mathcal{R}^J} p(\mathcal{D}|\mathcal{M}, \theta, \eta) p(\eta|\mathcal{M}, \theta) p(\theta|\mathcal{M}) d\theta d\eta$

- Integrated likelihood, with integration over $\beta$ and $\eta$:
  $p(\mathcal{D}|\mathcal{M}, \sigma_y^2, \sigma_\eta^2) = \int_{\mathcal{R}^{J+d}} p(\mathcal{D}|\mathcal{M}, \eta, \beta, \sigma_y^2, \sigma_\eta^2) p(\eta|\mathcal{M}, \sigma_\eta^2) p(\beta|\mathcal{M}) d\beta d\eta$

Finally, we introduce a more general multilevel model, in which any regression coefficient may also vary by group, with a higher-level model for that variability, as opposed to just an intercept term. This model is:

$$\mathcal{M}: \quad y_{ij} = \beta^T x_{ij} + \eta_j^T z_{ij} + \epsilon_{ij}, \; \epsilon_{ij} \sim \mathcal{N}(0, \sigma_y^2), \; \eta_j \sim \mathcal{N}(0, \Sigma_\eta(v))$$
$$\beta \sim \mathcal{N}(\mu, \Sigma), \; \sigma_y^2 \sim P_{\sigma_y^2}, \; v \sim P_v \tag{4}$$

A key distinction here, which differentiates this from a linear model, is that $\Sigma_\eta$ is an inherent variable of the model, rather than a fixed Bayesian hyperparameter. Also, as $\Sigma_\eta$ is a symmetric positive-definite matrix, we parameterise this through a vector $v$ instead of specifying the full matrix, but we do not make any assumption about the form of $\Sigma_\eta$ beyond this. As an example, it could be $\Sigma_\eta(v) = vI$, where $I$ is the identity matrix, though this assumption of independence is restrictive. As before, we centre the group-level coefficients $\eta_j$ by absorbing the 'average' into the $\beta$ coefficient and, therefore, there can be an overlap between the variables included in $z_{ij}$ and $x_{ij}$.

To implement this integrated likelihood approach under any MCMC sampling scheme, such as SMC or reversible-jump MCMC, it is useful to calculate beforehand all products and sums involving just the data $\mathcal{D}$ that are used within the log integrated likelihoods, which are defined in Eqs 6, 7 and 8. Then, the MCMC sampling scheme samples any variance components of the model, accepting or rejecting the proposed state by evaluating the log integrated likelihood at this state. For example, in the simple linear model, first compute terms like $\Sigma_{ij} x_{ij}$, then sample $\sigma_*^2 \sim P_{\sigma^2}$ and accept or reject $\sigma_*^2$ via the log integrated likelihood $\log p(\mathcal{D}|\mathcal{M}, \sigma_*^2)$ (Eq 6). Computations involved in the log integrated likelihood are all included in the accompanying code, and this is agnostic to the MCMC sampling scheme chosen. After completing the MCMC sampling, the model evidence can be estimated as appropriate [20–22].

We can compare two competing models for the data using the Bayes factor: $BF_{mn} = p(\mathcal{D}|\mathcal{M}_m)/p(\mathcal{D}|\mathcal{M}_n)$. These may, for instance, contain different subsets of the independent variables or have different prior beliefs for the hyperparameters, though the data $\mathcal{D}$ must remain fixed, i.e. both models contain the same $n$ individuals. The value of the Bayes factor indicates the strength of evidence for one model over the other. Interpretation is generally provided via tables proposed by [23] or [24].

## Integrated likelihood for the linear model

The integrated likelihood for linear models can often be found in Bayesian textbooks, such as [8], though for clarity, we include this using the notation above. The integrated likelihood is:

$$
\begin{aligned}
p(\mathcal{D}|\mathcal{M}, \sigma^2) \quad &= \int_{\mathcal{R}^d} \frac{1}{(2\pi)^{d/2} |\Sigma|^{1/2}} \exp\left(-\frac{1}{2}(\beta - \mu)^T \Sigma^{-1}(\beta - \mu)\right) \times \\
&\qquad\qquad\qquad \prod_i \frac{1}{\sqrt{2\pi\sigma^2}} \exp\left(-\frac{1}{2\sigma^2}(y_i - \beta^T x_i)^2\right) d\beta \\
&= \frac{1}{(2\pi)^{(d+n)/2} |\Sigma|^{1/2} \sigma^n} \times \\
&\quad \int_{\mathcal{R}^d} \exp\left(-\frac{1}{2}\left((\beta - \mu)^T \Sigma^{-1}(\beta - \mu) + \frac{1}{\sigma^2}\sum_i (y_i - \beta^T x_i)^2\right)\right) d\beta
\end{aligned}
$$

Rearranging the integrand and integrating out $\beta$, we get:

$$p(\mathcal{D}|\mathcal{M}, \sigma^2) = \frac{|\tilde{\Sigma}|^{1/2}}{(2\pi\sigma^2)^{n/2}|\Sigma|^{1/2}} \exp\left(-\frac{1}{2}\left(\mu^T\Sigma^{-1}\mu + \frac{1}{\sigma^2}\sum_i y_i^2 - \tilde{\mu}^T\tilde{\Sigma}^{-1}\tilde{\mu}\right)\right)$$

where we define:

$$\tilde{\Sigma}^{-1}(\sigma^2) = \Sigma^{-1} + \frac{1}{\sigma^2}\sum_i x_i x_i^T, \quad \tilde{\mu}(\sigma^2) = \tilde{\Sigma}\left(\Sigma^{-1}\mu + \frac{1}{\sigma^2}\sum_i x_i y_i\right) \tag{5}$$

In practice, we work with the logarithm of the integrated likelihood, particularly for computational reasons. This is:

$$\log p(\mathcal{D}|\mathcal{M}, \sigma^2) = -\frac{1}{2}\left(\log|\tilde{\Sigma}^{-1}| + \log|\Sigma| + n\log(2\pi\sigma^2) + \mu^T\Sigma^{-1}\mu + \frac{1}{\sigma^2}\sum_i y_i^2 - \tilde{\mu}^T\tilde{\Sigma}^{-1}\tilde{\mu}\right) \tag{6}$$

## Integrated likelihood for the linear model with normal-inverse-gamma conjugate prior

For a linear model with conjugate normal-inverse-gamma prior $\mathcal{NIG}(a, b, 0, \Sigma)$, we define the following:

$$\hat{\Sigma}^{-1} = \Sigma^{-1} + \sum_{i,j} x_{ij} x_{ij}^T, \quad \hat{\mu} = (\Sigma^{-1} + \sum_{i,j} x_{ij} x_{ij}^T)^{-1}\sum_{i,j} x_{ij} y_{ij}$$

$$b' = b + \frac{1}{2}\left(\sum_{i,j} y_{ij}^2 - (\sum_{i,j} x_{ij} y_{ij})^T(\Sigma^{-1} + \sum_{i,j} x_{ij} x_{ij}^T)^{-1}(\sum_{i,j} x_{ij} y_{ij})\right)$$

The posterior distribution for $\beta$ and $\sigma^2$ is then also normal-inverse-gamma, $\mathcal{NIG}(n/2 + a, b', \hat{\mu}, \hat{\Sigma})$. The log integrated likelihood and full log model evidence are (see [8], but note this uses a different reparameterisation):

$$\log p(\mathcal{D}|\mathcal{M}, \sigma^2) = -\frac{1}{2}\left(\log|\tilde{\Sigma}^{-1}| + \log|\Sigma| + n\log(2\pi\sigma^2)\right.$$

$$\left. + \frac{1}{\sigma^2}\mu^T\Sigma^{-1}\mu + \frac{1}{\sigma^2}\sum_i y_i^2 - \frac{1}{\sigma^2}\tilde{\mu}^T\tilde{\Sigma}^{-1}\tilde{\mu}\right)$$

$$\log p(\mathcal{D}|\mathcal{M}) = -\frac{1}{2}\left(\log|\hat{\Sigma}^{-1}| + \log|\Sigma| + n\log(2\pi) - 2a\log b + (2a + n)\log(b')\right.$$

$$\left. - 2\log\Gamma(n/2 + a) + 2\log\Gamma(a)\right)$$

## Integrated likelihood for a simple multilevel linear model

For the simple multilevel linear model (Eq 3):

$$
\begin{aligned}
p(\mathcal{D}|\mathcal{M}, \sigma_y^2, \sigma_\eta^2) \quad &= \int_{\mathcal{R}^d} \int_{\mathcal{R}^J} \frac{1}{(2\pi)^{d/2}|\Sigma|^{1/2}} \exp\left(-\frac{1}{2}(\beta - \mu)^T \Sigma^{-1}(\beta - \mu)\right) \times \\
&\qquad \prod_j \frac{1}{\sqrt{2\pi\sigma_\eta^2}} \exp\left(-\frac{\eta_j^2}{2\sigma_\eta^2}\right) \times \\
&\qquad \prod_{i,j} \frac{1}{\sqrt{2\pi\sigma_y^2}} \exp\left(-\frac{(y_{ij} - \beta^T x_{ij} - \eta_j)^2}{2\sigma_y^2}\right) d\eta\, d\beta \\
&= \frac{1}{\sigma_y^n |\Sigma|^{1/2} (2\pi)^{(n+d)/2}} \times \\
&\qquad \int_{\mathcal{R}^d} \exp\left(-\frac{1}{2}(\beta - \mu)^T \Sigma^{-1}(\beta - \mu)\right) \times \\
&\qquad \prod_j \left[ \frac{1}{\sqrt{2\pi\sigma_\eta^2}} \int_{-\infty}^{\infty} \exp\left(-\frac{\eta_j^2}{2\sigma_\eta^2} - \frac{1}{2\sigma_y^2} \sum_i (y_{ij} - \beta^T x_{ij} - \eta_j)^2\right) d\eta_j \right] d\beta
\end{aligned}
$$

Note that

$$
\prod_{i,j} \frac{1}{\sqrt{2\pi\sigma_y^2}} = (2\pi\sigma_y^2)^{-\sum_j n_j/2} = (2\pi\sigma_y^2)^{-n/2}
$$

We first consider the integral in square brackets, completing the square in $\eta_j$ in the expression:

$$
\begin{aligned}
\frac{\eta_j^2}{\sigma_\eta^2} + \frac{1}{\sigma_y^2} \sum_i (y_{ij} - \beta^T x_{ij} - \eta_j)^2 \quad &= \quad \frac{\sigma_y^2 + n_j \sigma_\eta^2}{\sigma_y^2 \sigma_\eta^2} \left( \eta_j - \frac{\sigma_\eta^2}{\sigma_y^2 + n_j \sigma_\eta^2} \sum_i (y_{ij} - \beta^T x_{ij}) \right)^2 \\
&\qquad + \frac{1}{\sigma_y^2} \sum_i (y_{ij} - \beta^T x_{ij})^2 \\
&\qquad - \frac{1}{\sigma_y^2} \frac{\sigma_\eta^2}{\sigma_y^2 + n_j \sigma_\eta^2} \left( \sum_i (y_{ij} - \beta^T x_{ij}) \right)^2
\end{aligned}
$$

This gives:

$$
\begin{aligned}
\frac{1}{\sqrt{2\pi\sigma_\eta^2}} &\int_{-\infty}^{\infty} \exp\left(-\frac{\eta_j^2}{2\sigma_\eta^2} - \frac{1}{2\sigma_y^2} \sum_i (y_{ij} - \beta^T x_{ij} - \eta_j)^2\right) d\eta_j \\
&= \sqrt{\frac{\sigma_y^2}{\sigma_y^2 + n_j \sigma_\eta^2}} \exp\left(-\frac{1}{2\sigma_y^2} \left( \sum_i (y_{ij} - \beta^T x_{ij})^2 - \frac{\sigma_\eta^2}{\sigma_y^2 + n_j \sigma_\eta^2} \left( \sum_i (y_{ij} - \beta^T x_{ij}) \right)^2 \right)\right)
\end{aligned}
$$

Then, rearranging for $\beta$ as in the linear model case:

$$
\begin{aligned}
(\beta - \mu)^T \Sigma^{-1}(\beta - \mu) &+ \frac{1}{\sigma_y^2} \sum_{i,j} (y_{ij} - \beta^T x_{ij})^2 - \frac{1}{\sigma_y^2} \sum_j \left( \frac{\sigma_\eta^2}{\sigma_y^2 + n_j \sigma_\eta^2} \left( \sum_i (y_{ij} - \beta^T x_{ij}) \right)^2 \right) \\
&= (\beta - \hat{\mu})^T \hat{\Sigma}^{-1}(\beta - \hat{\mu}) + \mu^T \Sigma^{-1} \mu + \frac{1}{\sigma_y^2} \sum_{i,j} y_{ij}^2 - \frac{1}{\sigma_y^2} \sum_j \left( \frac{\sigma_\eta^2}{\sigma_y^2 + n_j \sigma_\eta^2} \left( \sum_i y_{ij} \right)^2 \right) - \hat{\mu}^T \hat{\Sigma}^{-1} \hat{\mu}
\end{aligned}
$$

where we define:

$$\hat{\Sigma}^{-1}(\sigma_y^2, \sigma_\eta^2) = \Sigma^{-1} + \frac{1}{\sigma_y^2}\sum_{i,j}x_{ij}x_{ij}^T - \frac{1}{\sigma_y^2}\sum_j\left(\frac{\sigma_\eta^2}{\sigma_y^2 + n_j\sigma_\eta^2}(\sum_i x_{ij})(\sum_k x_{kj}^T)\right)$$

$$\hat{\mu}(\sigma_y^2, \sigma_\eta^2) = \hat{\Sigma}\left(\Sigma^{-1}\mu + \frac{1}{\sigma_y^2}\sum_{i,j}x_{ij}y_{ij} - \frac{1}{\sigma_y^2}\sum_j\left(\frac{\sigma_\eta^2}{\sigma_y^2 + n_j\sigma_\eta^2}(\sum_i y_{ij})(\sum_k x_{kj})\right)\right)$$

Finally, we get the integrated likelihood for the simple multilevel linear model:

$$p(\mathcal{D}|\mathcal{M}, \sigma_y^2, \sigma_\eta^2) = \frac{|\hat{\Sigma}|^{1/2}}{(2\pi\sigma_y^2)^{n/2}|\Sigma|^{1/2}}\prod_j\left(\sqrt{\frac{\sigma_y^2}{\sigma_y^2 + n_j\sigma_\eta^2}}\right)\times$$
$$\exp\left(-\frac{1}{2}\left(\mu^T\Sigma^{-1}\mu + \frac{1}{\sigma_y^2}\sum_{i,j}y_{ij}^2\right.\right.$$
$$\left.\left.-\frac{1}{\sigma_y^2}\sum_j\left(\frac{\sigma_\eta^2}{\sigma_y^2 + n_j\sigma_\eta^2}(\sum_i y_{ij})^2\right) - \hat{\mu}^T\hat{\Sigma}^{-1}\hat{\mu}\right)\right)$$

As before, a version of this integrated likelihood derivation can also be found in [8], but, in this case, it is given in simplified matrix algebra form, where the dependence on $\sigma_y^2$ and $\sigma_\eta^2$ is left unspecified. The log integrated likelihood is:

$$\log p(\mathcal{D}|\mathcal{M}, \sigma_y^2, \sigma_\eta^2) = -\frac{1}{2}\left(\log|\hat{\Sigma}^{-1}| + \log|\Sigma| + n\log(2\pi\sigma_y^2) + \sum_j\log\left(\frac{\sigma_y^2 + n_j\sigma_\eta^2}{\sigma_y^2}\right)\right.$$
$$+\mu^T\Sigma^{-1}\mu + \frac{1}{\sigma_y^2}\sum_{i,j}y_{ij}^2 - \frac{1}{\sigma_y^2}\sum_j\left(\frac{\sigma_\eta^2}{\sigma_y^2 + n_j\sigma_\eta^2}(\sum_i y_{ij})^2\right) \qquad (7)$$
$$\left.-\hat{\mu}^T\hat{\Sigma}^{-1}\hat{\mu}\right)$$

## Integrated likelihood for a general multilevel linear model

In the more general case (Eq 4), the steps are almost identical:

$$p(\mathcal{D}|\mathcal{M}, \sigma_y^2, v) = \int_{\mathcal{R}^d}\int_{\mathcal{R}^{mJ}}\frac{1}{(2\pi)^{d/2}|\Sigma|^{1/2}}\exp\left(-\frac{1}{2}(\beta - \mu)^T\Sigma^{-1}(\beta - \mu)\right)\times$$
$$\prod_j\frac{1}{(2\pi)^{m/2}|\Sigma_\eta(v)|^{1/2}}\exp\left(-\frac{1}{2}\eta_j^T\Sigma_\eta^{-1}(v)\eta_j\right)\times$$
$$\prod_{i,j}\frac{1}{\sqrt{2\pi\sigma_y^2}}\exp\left(-\frac{(y_{ij} - \beta^Tx_{ij} - \eta_j^Tz_{ij})^2}{2\sigma_y^2}\right)d\eta d\beta$$
$$= \frac{1}{\sigma_y^n|\Sigma|^{1/2}|\Sigma_\eta(v)|^{J/2}(2\pi)^{(n+d+mJ)/2}}\times$$
$$\int_{\mathcal{R}^{d+mJ}}\exp\left(-\frac{1}{2}(\beta - \mu)^T\Sigma^{-1}(\beta - \mu)\right)\times$$
$$\exp\left(-\frac{1}{2}\sum_j\eta_j^T\Sigma_\eta^{-1}(v)\eta_j - \frac{1}{2\sigma_y^2}\sum_i(y_{ij} - \beta^Tx_{ij} - \eta_j^Tz_{ij})^2\right)d\eta d\beta$$

Then:

$$(\beta - \mu)^T \Sigma^{-1} (\beta - \mu) + \sum_j \eta_j^T \Sigma_\eta^{-1} \eta_j + \frac{1}{\sigma_y^2} \sum_{i,j} (y_{ij} - \beta^T x_{ij} - \eta_j^T z_{ij})^2$$

$$= (\beta - \mu)^T \Sigma^{-1} (\beta - \mu) + \sum_j \left( \eta_j^T \left( \Sigma_\eta^{-1} + \frac{1}{\sigma_y^2} \sum_i z_{ij} z_{ij}^T \right) \eta_j - \eta_j^T \left( \frac{1}{\sigma_y^2} \sum_i z_{ij} (y_{ij} - \beta^T x_{ij}) \right) \right.$$

$$\left. - \left( \frac{1}{\sigma_y^2} \sum_i (y_{ij} - \beta^T x_{ij}) z_{ij}^T \right) \eta_j + \frac{1}{\sigma_y^2} \sum_i (y_{ij} - \beta^T x_{ij})^2 \right)$$

$$= (\beta - \mu)^T \Sigma^{-1} (\beta - \mu) + \sum_j \left( (\eta_j - \hat{\mu}_{\eta,j})^T \hat{\Sigma}_{\eta,j}^{-1} (\eta_j - \hat{\mu}_{\eta,j}) \right.$$

$$\left. + \frac{1}{\sigma_y^2} \sum_i (y_{ij} - \beta^T x_{ij})^2 - \hat{\mu}_{\eta,j}^T \hat{\Sigma}_{\eta,j}^{-1} \hat{\mu}_{\eta,j} \right)$$

$$= (\beta - \hat{\mu})^T \hat{\Sigma}^{-1} (\beta - \hat{\mu}) + \sum_j \left( (\eta_j - \hat{\mu}_{\eta,j})^T \hat{\Sigma}_{\eta,j}^{-1} (\eta_j - \hat{\mu}_{\eta,j}) \right) + \frac{1}{\sigma_y^2} \sum_{i,j} y_{ij}^2$$

$$- \frac{1}{\sigma_y^4} \sum_j \left( (\sum_i z_{ij}^T y_{ij}) \hat{\Sigma}_{\eta,j} (\sum_k z_{kj} y_{kj}) \right) + \mu^T \Sigma^{-1} \mu - \hat{\mu}^T \hat{\Sigma}^{-1} \hat{\mu}$$

where we now have additional definitions:

$$\hat{\Sigma}_{\eta,j}^{-1}(\sigma_y^2, v) = \Sigma_\eta^{-1}(v) + \frac{1}{\sigma_y^2} \sum_i z_{ij} z_{ij}^T, \qquad \hat{\mu}_{\eta,j}(\sigma_y^2, v) = \hat{\Sigma}_{\eta,j} \left( \frac{1}{\sigma_y^2} \sum_i z_{ij} (y_{ij} - \beta^T x_{ij}) \right)$$

$$\hat{\Sigma}^{-1}(\sigma_y^2, v) = \Sigma^{-1} + \frac{1}{\sigma_y^2} \sum_{i,j} x_{ij} x_{ij}^T - \frac{1}{\sigma_y^4} \sum_j \left( (\sum_i x_{ij} z_{ij}^T) \hat{\Sigma}_{\eta,j} (\sum_k z_{kj} x_{kj}^T) \right)$$

$$\hat{\mu}(\sigma_y^2, v) = \hat{\Sigma} \left( \Sigma^{-1} \mu + \frac{1}{\sigma_y^2} \sum_{i,j} x_{ij} y_{ij} - \frac{1}{\sigma_y^4} \sum_j \left( (\sum_i x_{ij} z_{ij}^T) \hat{\Sigma}_{\eta,j} (\sum_k z_{kj} y_{kj}) \right) \right)$$

Finally, we get the log integrated likelihood for the more general multilevel linear model:

$$\log p(\mathcal{D} | \mathcal{M}, \sigma_y^2, v) = -\frac{1}{2} \left( \log|\hat{\Sigma}^{-1}| + \log|\Sigma| + n \log(2\pi \sigma_y^2) + J \log|\Sigma_\eta| \right.$$

$$+ \sum_j \log|\hat{\Sigma}_{\eta,j}^{-1}| + \mu^T \Sigma^{-1} \mu + \frac{1}{\sigma_y^2} \sum_{i,j} y_{ij}^2 \qquad (8)$$

$$\left. - \frac{1}{\sigma_y^4} \sum_j \left( (\sum_i z_{ij}^T y_{ij}) \hat{\Sigma}_{\eta,j} (\sum_k z_{kj} y_{kj}) \right) - \hat{\mu}^T \hat{\Sigma}^{-1} \hat{\mu} \right)$$

## Example: Simulation study

We illustrate this approach first on simulated datasets based on the Prophet model of [25], which seeks to model a variable $y$ as a non-linear function of time $t$. By specifying a suitable flexible multi-dimensional transform of $t$, which includes piece-wise linear and Fourier

transform terms, we convert this problem to a linear model (or by extension a multilevel model). The piece-wise linear component requires pre-specified points $s_n$, $n = 1, \ldots, d_1$, at which the function is continuous but not smooth, i.e. the gradient changes. The Fourier transform component requires a specified periodicity $P$ and is truncated at $2d_2$ terms. The dimension of $x$ is then $d = 1 + d_1 + 2d_2$. The basic model structure is then as follows, where $E[\cdot]$ denotes expectation:

$$E[y] = f_1(t) + f_2(t) = \beta^T x = \beta^T g(t)$$

$$f_1(t) = \lambda + \sum_{n=1}^{d_1} \delta_n(t - s_n)1_{\{t > s_n\}}, \quad f_2(t) = \sum_{n=1}^{d_2}(a_n \cos(2\pi n t/P) + b_n \sin(2\pi n t/P))$$

$$\beta^T = (\lambda, \delta^T, a^T, b^T), \quad x^T = (1, (t - s_1)1_{\{t > s_1\}}, \ldots, \cos(2\pi t/P), \ldots, \sin(2\pi d_2 t/P)) = g(t)^T$$

Though this describes a non-linear relationship between $y$ and $t$, the model itself is linear because it is linear in the coefficient $\beta$. For each model type we have described in previous sections (linear, simple multilevel, general multilevel and linear model with fully conjugate normal-inverse-gamma prior), we generate a simulated dataset corresponding to that model, i.e. as if this is the 'true' model. We then evaluate each model on all four datasets, estimating the model evidence via the integrated likelihood and the full likelihood.

To generate the data, we first simulate multilevel group structure and $t_{ij}$, which in turn generates covariates $x_{ij}$ that have the form above. The covariates $z_{ij}$, which are those that vary by group within a general multilevel model, are defined as a (centred) subset of the $x_{ij}$. Both group structure and covariates, including $z_{ij}$, are shared across all datasets. The underlying structure is not explicitly used in the linear model or the associated linear dataset, as there is no relationship between the group membership and outcome variables. For each dataset, we sample 'true' model coefficients, which are then regarded as fixed, and we then compute the outcome variable, $y_{ij}$, as defined by the form of the corresponding model. Together with the multilevel structure and the covariates, this outcome variable forms the dataset, $\mathcal{D} = (y_{ij}, x_{ij}, z_{ij})$. We describe the datasets in more detail below.

In all datasets, we set $J = 15$ and $n = 1000$, which are the number of groups and of observations respectively. To assign multilevel group membership within the data, we sample the integers $j = 1, \ldots, J$ with replacement with probability $p_j$. In order to generate unequal group sizes, we sample $p_j$ from a Dirichlet distribution with parameter $\alpha = (2, \ldots, J + 1)$, such that $\Sigma_j p_j = 1$ and that $E[p_j] = (j + 1)/J^2$. We also have:

- For all datasets:

$$t_{ij} \sim U[0, 1]$$
$$d_1 = 5, \quad s = (0, 0.2, 0.4, 0.6, 0.8), \quad P = 1, \quad d_2 = 20, \quad d = 46$$
$$x_{ij}^T = (1, t_{ij}, (t_{ij} - 0.2)1_{\{t_{ij} > 0.2\}}, \ldots, \cos(2\pi t_{ij}), \ldots, \sin(6\pi t_{ij}))$$
$$z_{ij}^T = (1, t_{ij} - 0.5, (t_{ij} - 0.4)1_{\{t_{ij} > 0.4\}} - 0.18, (t_{ij} - 0.8)1_{\{t_{ij} > 0.8\}} - 0.02)$$

The constants added to $z_{ij}$ are such that $E[z_{ij}] = (1, 0, 0)$. We also specify a covariance hyperparameter $S$ for simulating 'true' coefficients $b$ from a multivariate Gaussian distribution,

where $S$ is a $d \times d$ positive-definite:

$$S_1 = \begin{pmatrix} 1 & 0 & 0 & 0 & 0 & 0 \\ 0 & 4 & -3 & -1 & 0 & 0 \\ 0 & -3 & 5 & -4 & 2 & 0 \\ 0 & -1 & -4 & 10 & -4 & 0 \\ 0 & 0 & 2 & -4 & 5 & 2 \\ 0 & 0 & 0 & 0 & 2 & 6 \end{pmatrix}, \ \lambda = 0.001, \ S = \begin{pmatrix} S_1 & 0 \\ 0 & \lambda I \end{pmatrix}$$

The elements of $S_1$ were chosen to allow flexibility in the gradients of $t$ in each interval $[s_n, s_{n+1}]$, with larger values as $\delta_n$ represents only a change in the gradient from the interval $[s_{n-1}, s_n]$ to the interval $[s_n, s_{n+1}]$. Similarly, $\lambda$ was set to a small value so that the Fourier element did not dominate the piece-wise linear component.

- Linear dataset:

$$\mathcal{D}_0: \quad b \sim \mathcal{N}(0, S), \ s^2 \sim \mathcal{IG}(3, 0.4)$$
$$e_{ij} \sim \mathcal{N}(0, s^2), \ y_{ij}^{(0)} = b^T x_{ij} + e_{ij}, \ \mathcal{D}_0 = (y_{ij}^{(0)}, x_{ij}, z_{ij})$$

- Simple multilevel dataset:

$$\mathcal{D}_1: \quad b \sim \mathcal{N}(0, S), \ s_y^2 \sim \mathcal{IG}(3, 0.3), \ s_h^2 \sim \mathcal{IG}(3, 0.1)$$
$$e_{ij} \sim \mathcal{N}(0, s_y^2), \ h_j \sim \mathcal{N}(0, s_h^2), \ y_{ij}^{(1)} = b^T x_{ij} + h_j + e_{ij}, \ \mathcal{D}_1 = (y_{ij}^{(1)}, x_{ij}, z_{ij})$$

- General multilevel dataset:

$$\mathcal{D}_2: \quad b \sim \mathcal{N}(0, S), \ s_y^2 \sim \mathcal{IG}(3, 0.3), \ s_{h,1}^2, s_{h,2}^2, s_{h,3}^2, s_{h,4}^2 \sim \mathcal{IG}(3, 0.1), \ \rho = 0.2$$

$$e_{ij} \sim \mathcal{N}(0, s_y^2), \ h_j \sim \mathcal{N}(0, S_h), \ S_h = \begin{pmatrix} s_{h,1}^2 & 0 & 0 & 0 \\ 0 & s_{h,2}^2 & \rho s_{h,2} s_{h,3} & 0 \\ 0 & \rho s_{h,2} s_{h,3} & s_{h,3}^2 & \rho s_{h,3} s_{h,4} \\ 0 & 0 & \rho s_{h,3} s_{h,4} & s_{h,4}^2 \end{pmatrix}$$

$$y_{ij}^{(2)} = b^T x_{ij} + h_j^T z_{ij} + e_{ij}, \ \mathcal{D}_2 = (y_{ij}^{(2)}, x_{ij}, z_{ij})$$

- Linear dataset using normal-inverse-gamma distribution:

$$\mathcal{D}_3: \quad s^2 \sim \mathcal{IG}(3, 0.4), \ b|s^2 \sim \mathcal{N}(0, \gamma s^2 S), \ (\gamma = 5 = 1/E[s^2])$$
$$e_{ij} \sim \mathcal{N}(0, s^2), \ y_{ij}^{(3)} = b^T x_{ij} + e_{ij}, \ \mathcal{D}_3 = (y_{ij}^{(3)}, x_{ij}, z_{ij})$$

The joint distribution for $b$ and $s^2$ is $\mathcal{NIG}(3, 0.4, 0, 5S)$, which is a conjugate prior for the Gaussian linear model, with $E[\beta] = 0$, $\text{Cov}[\beta] = S$.

As the expected value of $\mathcal{IG}(a, b)$ is $b/(a - 1)$, and $\eta_j$ and $\epsilon_{ij}$ are independent, in each case the outcome variable $y_{ij}$ should have similar expected value and variance, if the entire data generation was repeated multiple times, i.e. $E_\theta[E[y_{ij}|\theta]] = 0$ and $E_\theta[\text{var}(y_{ij} - b^T x_{ij}|\theta)] = 0.2$, where $\theta$ here indicates both covariate values and coefficients, e.g. $\theta = (t_{ij}, b, s^2)$. This is important as it

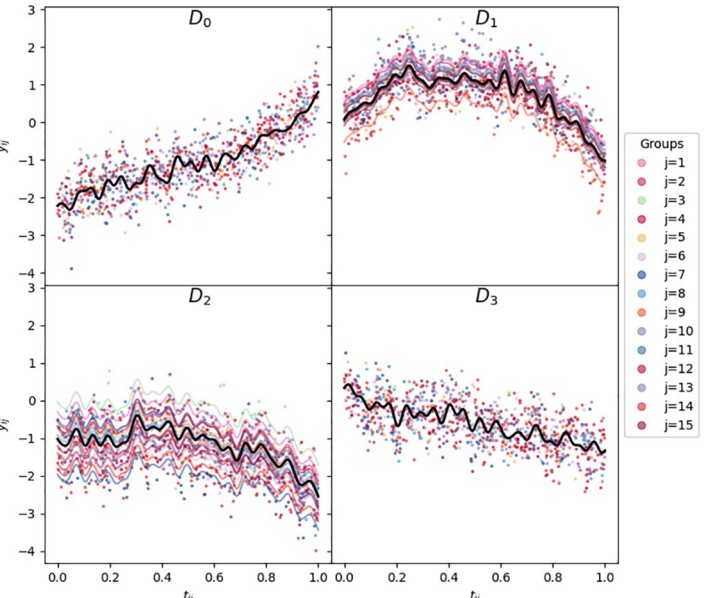

**Fig 1. Simulated datasets, $\mathcal{D}_0, \ldots, \mathcal{D}_3$.** Each dot represents a datapoint, with the model covariates $x = g(t)$ a deterministic and multi-valued non-linear function of $t$. In addition, the line $b^T x$ is shown for all values of $t \in [0, 1]$ for each dataset. For $\mathcal{D}_2$, the lines $b^T x + h_j$ are also included, for $j = 1, \ldots, J = 15$. Similarly, for $\mathcal{D}_3$, the lines $b^T x + h_j^T z$ are also included, where $z$ is also a deterministic multi-valued function of $t$.

means the choice of model priors should contribute less to the model evidence than the model structure, when we evaluate each model against each dataset. All four simulated datasets are shown in Fig 1 and available in the accompanying repository.

We now regard the 'true' coefficients, e.g. $b$ and $s$, as fixed but unknown, and we specify models corresponding to each dataset. We specify priors for each model, which are similar to the distributions from which we generated the 'true' coefficients. In place of the covariance matrix $S$ used to generate data, the prior for $\beta$ has covariance $\Sigma$, which shares the diagonal terms of $S$ but is zero elsewhere. The models are as follows:

- For all models:

$$\Sigma_1 = \begin{pmatrix} 1 & 0 & 0 & 0 & 0 & 0 \\ 0 & 4 & 0 & 0 & 0 & 0 \\ 0 & 0 & 5 & 0 & 0 & 0 \\ 0 & 0 & 0 & 10 & 0 & 0 \\ 0 & 0 & 0 & 0 & 5 & 0 \\ 0 & 0 & 0 & 0 & 0 & 6 \end{pmatrix}, \ \lambda = 0.001, \ \Sigma = \begin{pmatrix} \Sigma_1 & 0 \\ 0 & \lambda I \end{pmatrix}$$

- Linear model:

$$\mathcal{M}_0 : \quad y_i = \beta^T x_i + \epsilon_i, \ \epsilon_i \sim \mathcal{N}(0, \sigma^2)$$
$$\beta \sim \mathcal{N}(0, \Sigma), \ \sigma^2 \sim \mathcal{IG}(3, 0.4)$$

- Simple multilevel model:

$$\mathcal{M}_1: \quad y_{ij} = \beta^T x_{ij} + \eta_j + \epsilon_{ij}, \ \epsilon_{ij} \sim \mathcal{N}(0, \sigma_y^2), \ \eta_j \sim \mathcal{N}(0, \sigma_\eta^2)$$
$$\beta \sim \mathcal{N}(0, \Sigma), \ \sigma_y^2 \sim \mathcal{IG}(3, 0.4) \ , \sigma_\eta^2 \sim \mathcal{IG}(3, 0.1)$$

- General multilevel model:

$$\mathcal{M}_2: \quad y_{ij} = \beta^T x_{ij} + \eta_j^T z_{ij} + \epsilon_{ij}, \ \epsilon_{ij} \sim \mathcal{N}(0, \sigma_y^2), \ \eta_j \sim \mathcal{N}(0, \Sigma_\eta)$$
$$\beta \sim \mathcal{N}(0, \Sigma), \ \sigma_y^2 \sim \mathcal{IG}(3, 0.3), \ \sigma_{v,1}^2, \sigma_{v,2}^2, \sigma_{v,3}^2 \sim \mathcal{IG}(3, 0.1), \ \rho = 0.2$$
$$\Sigma_\eta = \begin{pmatrix} \sigma_{v,1}^2 & 0 & 0 & 0 \\ 0 & \sigma_{v,2}^2 & \rho\sigma_{v,2}\sigma_{v,3} & 0 \\ 0 & \rho\sigma_{v,2}\sigma_{v,3} & \sigma_{v,3}^2 & \rho\sigma_{v,3}\sigma_{v,4} \\ 0 & 0 & \rho\sigma_{v,3}\sigma_{v,4} & \sigma_{v,4}^2 \end{pmatrix}$$

- Linear model using normal-inverse-gamma distribution prior:

$$\mathcal{M}_3: \quad y_i = \beta^T x_i + \epsilon_i, \ \epsilon_i \sim \mathcal{N}(0, \sigma^2)$$
$$\sigma^2 \sim \mathcal{IG}(3, 0.4), \ \gamma = 5, \ \beta | \sigma^2 \sim \mathcal{N}(0, \gamma\sigma^2\Sigma)$$

For each dataset, we expect the model with 'true' structure to have the largest model evidence. Table 1 shows the performance of the models on the respective datasets, illustrating how our approach not only leads to a decrease in the uncertainty of the estimated model evidence, but can also prevent model misspecification where sampling using the full likelihood is unable to do so, correctly identifying the 'true' model for each dataset. For models $\mathcal{M}_0$ to $\mathcal{M}_2$, we cannot evaluate the bias of the log model evidence estimates, with no direct solution to compare to. However, for the linear model with normal-inverse-gamma prior, direct computation shows that estimates using the integrated likelihood are unbiased and closer to the

**Table 1. Comparison of model evidence for each simulated dataset and model.** This was computed with sequential Monte Carlo (SMC), using the integrated likelihood and separately using the full likelihood. This was repeated for 8 random initialisations with 2000 draws at each step in SMC, and we present the mean and standard deviation of the model evidence from each run. For each approach, the model with the strongest evidence is marked with * (this is not clear for $\mathcal{D}_3$ with rounding here, but see the repository [17] for full results). For each model, we also report the time taken to complete the sampling for all initialisations, averaged across the different datasets.

| | SMC, integrated likelihood | | | | |
| --- | --- | --- | --- | --- | --- |
| | $\log p(\mathcal{D}_0|\mathcal{M})$ | $\log p(\mathcal{D}_1|\mathcal{M})$ | $\log p(\mathcal{D}_2|\mathcal{M})$ | $\log p(\mathcal{D}_3|\mathcal{M})$ | Time, s |
| $\mathcal{M}_0$ | -633.08 (0.03)* | -753.53 (0.05) | -908.24 (0.07) | -684.87 (0.05) | 48.2 |
| $\mathcal{M}_1$ | -642.08 (0.04) | -681.06 (0.02)* | -518.24 (0.04) | -694.88 (0.06) | 69.5 |
| $\mathcal{M}_2$ | -644.66 (0.05) | -683.43 (0.03) | -508.96 (0.06)* | -697.33 (0.06) | 298.5 |
| $\mathcal{M}_3$ | -633.10 (0.05) | -753.50 (0.04) | -909.22 (0.03) | -684.87 (0.03)* | 17.4 |
| | SMC, full likelihood | | | | |
| | $\log p(\mathcal{D}_0|\mathcal{M})$ | $\log p(\mathcal{D}_1|\mathcal{M})$ | $\log p(\mathcal{D}_2|\mathcal{M})$ | $\log p(\mathcal{D}_3|\mathcal{M})$ | Time, s |
| $\mathcal{M}_0$ | -633.34 (0.15) | -753.79 (0.18) | -908.29 (0.22) | -685.00 (0.22) | 110.1 |
| $\mathcal{M}_1$ | -643.23 (0.24) | -681.37 (0.32)* | -526.05 (2.69)* | -695.64 (0.26) | 152.8 |
| $\mathcal{M}_2$ | -647.68 (1.07) | -686.42 (0.99) | -532.39 (6.75) | -700.39 (1.05) | 505.7 |
| $\mathcal{M}_3$ | -632.95 (0.26)* | -753.56 (0.19) | -909.44 (0.18) | -684.63 (0.21)* | 362.8 |
| | Fully analytical solution | | | | |
| | $\log p(\mathcal{D}_0|\mathcal{M})$ | $\log p(\mathcal{D}_1|\mathcal{M})$ | $\log p(\mathcal{D}_2|\mathcal{M})$ | $\log p(\mathcal{D}_3|\mathcal{M})$ | |
| $\mathcal{M}_3$ | -633.08 | -753.47 | -909.23 | -684.87 | |

**Table 2. Comparison of Mahalabonis distance between 'true' coefficient *b* and model posterior for *β*, for each simulated datasets and model.** This was computed with sequential Monte Carlo (SMC), separately using the integrated likelihood and the full likelihood.

| | Integrated: $d_{M,i}(b, P(\beta|\mathcal{D}, \mathcal{M}))$ | | | | Full: $d_{M,f}(b, P(\beta|\mathcal{D}, \mathcal{M}))$ | | | |
|---|---|---|---|---|---|---|---|---|
| | $\mathcal{D}_0$, $b$ | $\mathcal{D}_1$, $b$ | $\mathcal{D}_2$, $b$ | $\mathcal{D}_3$, $b$ | $\mathcal{D}_0$, $b$ | $\mathcal{D}_1$, $b$ | $\mathcal{D}_2$, $b$ | $\mathcal{D}_3$, $b$ |
| $\mathcal{M}_0$, $P(\beta|\mathcal{D}, \mathcal{M}_0)$ | 7.64 | 6.65 | 13.74 | 6.99 | 7.78 | 6.73 | 13.90 | 7.05 |
| $\mathcal{M}_1$, $P(\beta|\mathcal{D}, \mathcal{M}_1)$ | 7.68 | 6.55 | 6.55 | 6.81 | 7.88 | 6.63 | 6.76 | 6.97 |
| $\mathcal{M}_2$, $P(\beta|\mathcal{D}, \mathcal{M}_2)$ | 7.68 | 6.53 | 6.36 | 6.67 | 8.61 | 7.39 | 7.18 | 7.41 |
| $\mathcal{M}_3$, $P(\beta|\mathcal{D}, \mathcal{M}_3)$ | 3.38 | 3.17 | 7.77 | 3.19 | 7.86 | 6.48 | 13.65 | 7.01 |

analytic solution than the estimates using the full likelihood. Furthermore, in every instance within this example the computational cost of running the SMC sampling algorithm on the variance parameters with the integrated likelihood was reduced compared to sampling all parameters with the full likelihood.

We can also compare the posterior distributions for model coefficients, compared to the 'true' coefficients. Table 2 shows the Mahalabonis distance between *b* and the posterior distribution for *β*, for each combination of dataset and model. In every case, the 'true' coefficient is closer to the posterior distribution from the MCMC using the integrated likelihood than it is to the posterior from the MCMC using the full likelihood. The Mahalabonis distance is defined as the following, where the posterior for *β* has mean *μ* and covariance Σ:

$$d_M(b, P(\beta|\mathcal{D}, \mathcal{M})) = \sqrt{(b - \mu)^T \Sigma^{-1} (b - \mu)}$$

For the integrated likelihoods, the posterior mean and covariance can be recovered by averaging $\tilde{\mu}$ and $\tilde{\Sigma}$ (or $\hat{\mu}$ and $\hat{\Sigma}$) over all values of $\sigma^2$ (or $\sigma_y^2$ and $\sigma_\eta^2$) in the MCMC posterior trace. For the full likelihoods, the posterior mean and covariance are computed directly as the sample mean and sample covariance from the MCMC trace for *β*. For example, for $\mathcal{M}_0$, the Mahalabonis distances for integrated and full likelihoods are as follows, where $\tilde{\Sigma}^{-1}(\sigma^2)$ and $\tilde{\mu}(\sigma^2)$ are the expressions in Eq 5:

$$\text{Integrated}: \quad d_{M,i}(b, P(\beta|\mathcal{D}, \mathcal{M}_0)) \quad = \sqrt{\frac{1}{N} \sum_n (b - \tilde{\mu}(\sigma_n^2))^T \tilde{\Sigma}^{-1}(\sigma_n^2)(b - \tilde{\mu}(\sigma_n^2))}$$

$$\text{Full}: \quad d_{M,f}(b, P(\beta|\mathcal{D}, \mathcal{M}_0)) \quad = \sqrt{(b - \bar{\beta})^T V^{-1} (b - \bar{\beta})}$$

$$\bar{\beta} = \frac{1}{N} \sum_n \beta_n, \quad V \quad = \frac{1}{N-1} \sum_n (\beta_n - \bar{\beta})(\beta_n - \bar{\beta})^T$$

## Example: Minnesota radon contamination

We next investigate real-world data, the *Minnesota radon contamination* dataset [19]. We describe various models that fit within this framework outlined above, as proposed for this dataset in [19]. We deviate from their notation (e.g. renaming coefficients) for consistency with our notation above. The hierarchical structure in this dataset is given in decreasing geographic granularity, with 919 individual measurements grouped within 85 counties, and data are available at the individual measurement-level and the county-level. The maximum number of measurements per county is 116 and the minimum is 1. The explanatory variable is the measurement of the radon level on a logarithmic scale, which has mean (standard deviation) 1.265Å (0.819). Comparing across counties, the minimum and maximum values for the

average log radon level was 0.410 and 2.606. The covariates considered here are an individual-level indicator variable identifying the floor the measurement was taken on (0 for basement, 1 for first floor), and county-wide uranium levels on a logarithmic scale. 83% of the measurements were taken in the basement, and the mean (standard deviation) of the log uranium levels was 0.014 (0.384), with a minimum and maximum across all counties of −0.882 and 0.528.

We denote the standardised (mean 0 and standard deviation 1) log radon measurements by $y_{ij}$, the indicator floor variable by $t_{ij}$, and the county-wide log uranium levels (also standardised) by $v_j$. Unless otherwise specified, we include an intercept term in each model, but adjust the model matrix $x_{ij}$ so that it becomes $x_{ij}^T = (1 - t_{ij}, t_{ij})$. We could equivalently rewrite this as $x_{ij}^T = (1_{\{t_{ij}=0\}}, 1_{\{t_{ij}=1\}})$, where the indicator function, $1_A$, is equal to 1 if condition $A$ is True and 0 otherwise. This means that we index by $t_{ij}$ rather than including it as a binary variable. The primary reason for this is that we then express the same prior uncertainty for measurements that come from the basement floor and from the first floor, instead of increased uncertainty when $t_{ij} = 1$, as discussed in [16]. In the context of linear model notation, we discard the group-level $j$ index, so that the index $i$ runs over all individuals, $i = 1, \ldots, n$. To denote the group $j$ ownership for a particular individual $i$, we instead using the index notation $j[i]$. For example, if the individual 10 belongs to group 4, then $j[10] = 4$. The models suggested by Gelman and Hill include the following single-level linear models:

- Complete pooling: all counties are pooled to a single group, with a single intercept and gradient used for all counties, whilst the county-wide uranium levels are not included in the model. By 'averaging' the intercept term, this completely ignores any variation in the radon levels across counties. The model is:

$$\mathcal{M}_0 : y_i = a + bt_i + \epsilon_i = \beta^T x_i + \epsilon_i, \ \beta^T = (a, b + a), \ x_i^T = (1 - t_i, t_i), \ \epsilon_i \sim \mathcal{N}(0, \sigma^2)$$

- Complete pooling, with county-level variables: as above, but with county-wide log uranium measurements included in the model. This at least contains some county-wide information, but does not directly model at the level of counties, as in the multilevel models.

$$\mathcal{M}_1 : \quad y_i = a + bt_i + cv_{j[i]} + \epsilon_i = \beta^T x_i + \epsilon_i$$
$$\beta^T = (a, b + a, c), \ x_i^T = (1 - t_i, t_i, v_{j[i]}), \ \epsilon_i \sim \mathcal{N}(0, \sigma^2)$$

- Unpooled intercept: each county has a separate intercept term. Although the county-level data is included via indicator variables that identify group membership, there is again no explicit model at the county-level. This is referred to as *no pooling* in the PyMC multilevel modelling notebook [16], though the coefficient for the floor/basement indicator variable is pooled across counties. We could also include the county-wide log uranium measurements here, but this will result in a non-identifiable model with collinear predictors.

$$\mathcal{M}_2 : \quad y_i = a_{j[i]} + a + bt_i + \epsilon_i = \beta^T x_i + \epsilon_i, \ \epsilon_i \sim \mathcal{N}(0, \sigma^2)$$
$$\beta^T = (a_1, \ldots, a_J, a, a + b), \ x_i^T = (1_{\{j[i]=1\}}, \ldots, 1_{\{j[i]=J\}}, (1 - t_i), t_i)$$

- No pooling: each county is modelled completely independently of others, with separate intercepts and gradients. This will usually overfit the data, and perform relatively poorly for counties with limited data. In practice, 25 out of 85 counties have no measurements from the first floor, and we exclude those components in the vectors $\beta$ and $x_i$. The dimension of $\beta$ is

then $85 + 60 = 145$.

$$\mathcal{M}_3: \quad y_i = a_{j[i]} + b_{j[i]}t_i + \epsilon_i = \beta^T x_i + \epsilon_i, \ \epsilon_i \sim \mathcal{N}(0, \sigma^2)$$
$$\beta^T = (a_1, \ldots, a_J, b_1 + a_1, \ldots, b_J + a_J)$$
$$x_i^T = (1_{\{j[i]=1\}}(1 - t_i), \ldots, 1_{\{j[i]=J\}}(1 - t_i), 1_{\{j[i]=1\}}t_i, \ldots, 1_{\{j[i]=J\}}t_i)$$

The multilevel models are the following:

- Partial pooling: county-wide variability is modelled directly as $\eta_j$, a deviation from the 'average' intercept. This uses first multilevel model formulation as described in (3).

$$\mathcal{M}_4: \quad y_{ij} = a + bt_{ij} + cv_j + \eta_j + \epsilon_{ij} = \beta^T x_{ij} + \eta_j + \epsilon_{ij}$$
$$\epsilon_{ij} \sim \mathcal{N}(0, \sigma_y^2), \ \eta_j \sim \mathcal{N}(0, \sigma_\eta^2)$$
$$\beta^T = (a, b + a, c), \ x_{ij}^T = (1 - t_{ij}, t_{ij}, v_j)$$

- Varying slopes and intercepts: in this model, we allow variability in both intercept and slope (i.e. the floor the measurement was taken on) across counties. This uses the more general multilevel model (described in Eq 4). We evaluate a version of this that includes an off-diagonal (correlation) term in the $\Sigma_\eta$ prior.

$$\mathcal{M}_5: \quad y_{ij} = a + bt_{ij} + cv_j + \eta_{j,1}(1 - t_{ij}) + \eta_{j,2}t_{ij} + \epsilon_{ij} = \beta^T x_{ij} + \eta_j^T z_{ij} + \epsilon_{ij}$$
$$\epsilon_{ij} \sim \mathcal{N}(0, \sigma_y^2), \ \eta_j \sim \mathcal{N}(0, \Sigma_\eta(v))$$
$$\beta^T = (a, b + a, c), \ x_{ij}^T = (1 - t_{ij}, t_{ij}, v_j), \ z_{ij}^T = (1 - t_{ij}, t_{ij})$$

Where complete pooling and no pooling represent two extremes in model dimension within the linear model framework, Gelman and Hill [19] describe the multilevel model as akin to *partial pooling*, in which there is natural shrinkage of the non-pooled parameters (e.g. those featuring the index $j[i]$) to the mean (the 'average' in the complete pooling case). This can be seen as a compromise between the two linear model extremes.

In each of these models, we set a multivariate normal prior $\mathcal{N}(0, I)$ on $\beta$ and inverse-gamma $\mathcal{IG}(3, 1)$ prior on each univariate variance component ($\sigma^2$, $\sigma_y^2$ and $\sigma_\eta^2$). In model $\mathcal{M}_5$, $\Sigma_\eta$ was parameterised by $v = (\sigma_{v,1}^2, \sigma_{v,2}^2, \rho_v)$, where the first two components were diagonal terms, which had $\mathcal{IG}(3, 1)$ priors, $\rho_v$ had a truncated normal prior on the interval $[-1, 1]$ with mean 0 and variance 1, and the non-zero off-diagonal term was $\rho_v \sigma_{v,1} \sigma_{v,2}$. The number of unconstrained model parameters, $k$, in linear models is equal to the number of independent variables, which is the same as the dimension of the model evidence integral. In multilevel models, integration also happens over latent variables, while $k$ is just number of independent variables plus the number of variance components. Table 3 compares these models, in terms of the model evidence and the AIC, where we use SMC to estimate the model evidence using the full likelihoods and the derived integrated likelihoods. S1–S6 Figs. shows measurements and model fits for a subset of counties.

## Discussion

Multilevel structure within data unlocks an increasing number of modelling choices for statisticians, though this additional modelling flexibility presents a challenge in deciding what and how to model the data. We present an approach to Bayesian model selection for multilevel models that estimates the model evidence using integrated likelihoods instead of full likelihoods. We treat a subset of variables (regression coefficients with Gaussian priors) as nuisance

**Table 3. Comparison of models for the *Minnesota radon contamination* dataset, using AIC and model evidence.** The model evidence was computed with sequential Monte Carlo (SMC), using separately the integrated likelihood presented and the full likelihood. This was repeated for 8 random initialisations with 2000 draws at each step in SMC, and we present the mean and standard deviation of the model evidence from each run. The table also shows the number of model parameters, $k$, and the ranking (where smaller is better) of models for each approach.

| | | | | SMC, integrated likelihood | | SMC, full likelihood | |
|---|---|---|---|---|---|---|---|
| Model | $k$ | AIC | rank | $\log p(\mathcal{D}|\mathcal{M})$ | rank | $\log p(\mathcal{D}|\mathcal{M})$ | rank |
| $\mathcal{M}_0$ | 2 | 2544.17 | 6 | -1279.87 (0.04) | 6 | -1279.85 (0.06) | 6 |
| $\mathcal{M}_1$ | 3 | 2427.74 | 3 | -1224.14 (0.05) | 1 | -1224.12 (0.07) | 1 |
| $\mathcal{M}_2$ | 86 | 2469.74 | 4 | -1263.61 (0.02) | 4 | -1261.31 (0.46) | 4 |
| $\mathcal{M}_3$ | 145 | 2496.63 | 5 | -1270.69 (0.05) | 5 | -1267.40 (1.70) | 5 |
| $\mathcal{M}_4$ | 5 | 2425.21 | 2 | -1226.93 (0.05) | 3 | -1231.11 (0.36) | 2 |
| $\mathcal{M}_5$ | 7 | 2423.11 | 1 | -1225.77 (0.03) | 2 | -1232.15 (1.75) | 3 |

variables that we analytically integrate out, which reduces the dimensionality of the model, as is standard in conjugate analysis. By converting the problem in this manner, we limit the impact of issues surrounding high-dimensional sampling, a key difficulty in sampling schemes for estimation of the desired quantities. As both examples show, estimates of the model evidence using the integrated likelihood are more consistent and robust. For the simulated data, this approach correctly identifies the 'true' model for each dataset and the 'true' coefficients are more closely described by the posterior distribution when using the integrated likelihood than when using the full likelihood. For a linear model with normal-inverse-gamma prior, we can also compute the log model evidence directly, and the estimates using the integrated likelihood have less bias and variance than those using the full likelihood. We believe the bias in the model evidence estimates using the full likelihood is likely shared by other models, particularly the multilevel models, because of higher dimensionality, though there is no gold standard to confirm this. These observations extend to the *Minnesota radon contamination* dataset, where the discrepancy between estimates and their variance is significant. The integrated likelihood is more consistent with the frequentist AIC, following a similar ranking, though this does not measure exactly the same thing. Although static SMC is asymptotically unbiased in the data size $n$ [20], it is sometimes unclear, when dealing with high-dimensional models, what constitutes an unbiased estimate in practice when there is no analytic solution available. In high-dimensional settings, methods that directly estimate the model evidence integral may easily accumulate errors, leading to poor estimates. In Table 1, we notice some improvement in computational cost for the simulated datasets, but we believe the computational cost depends on a number of factors, such as dimensions $n$, $d$ and $m$ and the number of groups $J$, and so are reluctant to make a general statement about this. Sampling using a highly-nonlinear low-dimensional integrated likelihood may in some instances be more computationally challenging than using the high-dimensional product of simpler likelihoods. In the second example, both full and integrated likelihood methods were broadly similar in terms of computational cost for the linear model and simple multilevel linear model, but the integrated likelihood was more expensive for the general multilevel model, as this involved repeated computation and inversion of a large number of covariance matrices.

Bayesian model selection can be extended to a wide range of related problems fairly straightforwardly, such as variable selection and nested models (i.e. a comparison of two models where one is entirely contained with the other, as opposed to a nested structure in the data). It is worth emphasising a distinction between the model that best describes the data and the model that best achieves the research objective, which may not always coincide. For example, if the goal is to make inference on parameters associated with specific variables, then we should not

exclude these variables on the basis of an evaluation of some model selection criteria. As George Box stated in one of the most well-known aphorisms in statistics [26]: 'All models are wrong, but some are useful'. In a model selection problem, Bayesian approaches are particularly advantageous, because they factor prior uncertainty about model parameters in a way that naturally imposes a penalty on model complexity to prevent overfitting to the data. An important consideration is the choice of suitable priors for a given problem, to adequately balance previous scientific knowledge and information that the new data provides. Informative or weakly informative priors are typically preferred to non-informative priors, which will often not be suitable if there is insufficient data available. For linear models, the Bayes factor is a monotonic function of the classical $F$ statistic in the limit as the prior variances tend to infinity [8]. Similarly, the posterior distribution of $\beta$ given $\sigma^2$ also aligns with classical frequentist inference in this limit. It is worth emphasising that *posteriori* maximisation of the model evidence over prior hyperparameters is generally not appropriate in the context of an inference question about particular dataset. Some caution should be taken to avoid 'retro-fitting' priors based on the data, as this can be viewed as converting *a priori* fixed hyperparameters (part of the model definition) into tunable parameters, under which the inference question may not remain as initially intended. However, empirical Bayes [27], which performs such an optimisation on a wider dataset (for example, using radon contamination from other states to estimate sensible priors), can be viewed as a shrinkage approach and a bridge from frequentist estimation to the fully Bayesian approach. With *a priori* justification, it is certainly possible to compare a discrete set of models that are identical except from different prior hyperparameters. For example, two statisticians may have wildly different prior beliefs based on previous research, and therefore propose separate prior distributions, which in turn can influence inference they make on model parameters, and we could ask whose model best describes the data. As always, prior predictive checking should be used to ensure priors give a reasonable coverage in predicted values.

We have limited this work to the simplest generalised linear models, with normal distribution and identity link (the canonical link function), choosing normal priors to mirror conjugate priors for a Gaussian likelihood. As any likelihood from the exponential family has a conjugate prior distribution, this analytic marginalisation can be similarly extended to generalised linear models under similarly chosen priors; for example, logistic regression (generalised linear model with Bernoulli distribution and logit link) with a beta conjugate prior on the Bernoulli parameter $p$. This allows the approach we have presented to be generalised to a much larger class of data and models.

## Data and source code availability

The source code and data is available in the following repository:

- Project name: Bayesian model selection for multilevel models using integrated likelihoods

- Project home page: https://github.com/tedinburgh/model-evidence-with-integrated-likelihood

- Operating system(s): Platform independent

- Programming language: Python 3.9.12

- Other requirements: Python modules—numpy 1.21.5 or higher, pandas 1.4.2 or higher, pmyc 4.1.5 or higher, arviz 0.12.1 or higher, scipy 1.7.3 or higher, argparse 1.1 or higher, statsmodels 0.13.2, palettable 3.3.0.

- License: MIT License

The current version of the repository has a permanent DOI at Zenodo [17]. The *Minnesota radon* dataset is contained within the module PyMC and can be opened directly from there and the simulated datasets can be reproduced exactly when running the relevant Python script from the repository above. Additionally, all datasets are available as.csv files on the repository above.

## Supporting information

**S1 Fig. Model $\mathcal{M}_0$ fits for a subset of counties.** Each dot represents a measurement at either basement or ground floor level. The format of the figure follows [16], with the same counties represented, though we have standardised the log radon and uranium levels, so the $y$-axis scale is slightly different. The model fit is from the integrated likelihood sampling, and is shown as a gradient line from basement to ground floor, with one standard deviation from the mean in dotted lines.
(TIF)

**S2 Fig. $\mathcal{M}_1$ model fits for subset of counties.**
(TIF)

**S3 Fig. $\mathcal{M}_2$ model fits for subset of counties.**
(TIF)

**S4 Fig. $\mathcal{M}_3$ model fits for subset of counties.**
(TIF)

**S5 Fig. $\mathcal{M}_4$ model fits for subset of counties.**
(TIF)

**S6 Fig. $\mathcal{M}_5$ model fits for subset of counties.**
(TIF)

## Acknowledgments

We would like to thank Dr Torben Sell (University of Edinburgh) for his insight and advice about challenges in MCMC sampling. A CC BY or equivalent licence is applied to the AAM arising from this submission.

## Author Contributions

**Conceptualization:** Tom Edinburgh.

**Methodology:** Tom Edinburgh.

**Software:** Tom Edinburgh.

**Supervision:** Ari Ercole, Stephen Eglen.

**Writing – original draft:** Tom Edinburgh.

**Writing – review & editing:** Tom Edinburgh, Ari Ercole, Stephen Eglen.

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
