## [Decision Letter · Decision Letter 0]

4 Oct 2022

PONE-D-22-21044Bayesian model selection for multilevel models using marginal likelihoodsPLOS ONE

Dear Dr. Edinburgh,

Thank you for submitting your manuscript to PLOS ONE. After careful consideration, we feel that it has merit but does not fully meet PLOS ONE’s publication criteria as it currently stands. Therefore, we invite you to submit a revised version of the manuscript that addresses the points raised during the review process. Please submit your revised manuscript by Nov 18 2022 11:59PM. If you will need more time than this to complete your revisions, please reply to this message or contact the journal office at plosone@plos.org. Please include the following items when submitting your revised manuscript:A rebuttal letter that responds to each point raised by the academic editor and reviewer(s). You should upload this letter as a separate file labeled 'Response to Reviewers'.A marked-up copy of your manuscript that highlights changes made to the original version. You should upload this as a separate file labeled 'Revised Manuscript with Track Changes'.An unmarked version of your revised paper without tracked changes. You should upload this as a separate file labeled 'Manuscript'.

We look forward to receiving your revised manuscript.

Kind regards,

Alessandro Barbiero, Ph.D. in Statistics

Academic Editor

PLOS ONE

Journal Requirements:

“We would like to thank Dr Torben Sell (University of Edinburgh) for his insight and advice about challenges in MCMC sampling. TE is funded by Engineering and Physical Sciences Research Council (EPSRC) National Productivity Investment Fund (NPIF) EP/S515334/1, reference 2089662. A CC BY or equivalent licence is applied to the AAM arising from this submission.”

“TE is funded by Engineering and Physical Sciences Research Council (EPSRC) National Productivity Investment Fund (NPIF) EP/S515334/1, reference 2089662. The funders had no role in study design, data collection and analysis, decision to publish, or preparation of the manuscript.

https://gow.epsrc.ukri.org/NGBOViewGrant.aspx?GrantRef=EP/S515334/1”

Additional Editor Comments:

I would suggest, as one of the referees does, that the authors include a simulation study to assess their proposition.

Reviewers' comments:

Reviewer's Responses to Questions

**Comments to the Author**

1. Is the manuscript technically sound, and do the data support the conclusions?

Reviewer #1: Partly

Reviewer #2: Yes

2. Has the statistical analysis been performed appropriately and rigorously? 

Reviewer #1: Yes

Reviewer #2: Yes

3. Have the authors made all data underlying the findings in their manuscript fully available?

Reviewer #1: Yes

Reviewer #2: Yes

4. Is the manuscript presented in an intelligible fashion and written in standard English?

Reviewer #1: Yes

Reviewer #2: Yes

5. Review Comments to the Author

Reviewer #1: Review on

“Bayesian model selection for multilevel models using marginal likelihoods”

This article proposes the use of marginal likelihoods instead of full likelihood to perform model selection for multilevel models. The methods are illustrated using generalized linear models where the responses are normally distributed, and the identity link is used. In this setting, the authors demonstrate that marginal likelihoods can be obtained analytically by integrating out the fixed and random effects. This reduces the dimensionality when computing the model evidence using SMC as only the variance parameters remain to be integrated out.

The manuscript is generally very well written with clear explanations although there are some typographical errors to be corrected. Below are some comments and suggestions for the authors’ consideration in revising the manuscript. A point-by-point response to these comments is appreciated.

1.I think using the term “marginal likelihood” to represent what is obtained after integrating a subset of the parameters is rather confusing because this term is usually taken to mean what is obtained after integrating out all the parameters. Perhaps “partial marginal likelihood” would be more accurate? Or the authors can consider other terms that provide a better description?

2.Page 5 line 151: Should $\\nu$ be $\\eta$ in “integrated over $\\beta$ and $\\nu$”?

3.Page 5 line 170: Typo in “include the this using the notation above”

4.Page 5 line 172: Typo in “Rearranging the square brackets” because there are no square brackets in the expression above this line.

5.Page 6 line 176: Since the product is over i and j, I wonder if the power for $2 \\pi \\sigma_y$ should be $nJ$ instead of just $n$? Same problem in equation (6).

6.Page 6 line 177: A \\sigma_\\eta^2$ is missing in the numerator of the very last term.

7.Page 6 line 179: Typo in “rearranging for with $\\beta$ as in the”

8.Page 7 line 186: In line 4 of the equation, the power of $|\\Sigma_\\eta(\\nu)|$ should be J instead of m. Similarly I wonder if the power of (2\\pi\\sigma_y) should be nJ instead of n? Same problem in equation (7).

9.There is only one example given on Minnesota radon contamination. Will the authors consider including a simulation study? For this real dataset, it is unclear how the models should be ranked since there is no true model.

10.There is a lack of discussion on the comparison of SMC(marginal likelihood) and SMC(full likelihood). It is not clear what are the advantages of SMC(marginal likelihood)? Is the computation time reduced? How does this reduction depend on the number of observations and parameters? If this is the case, the authors should present some results on the timings? What other advantages are there? The authors need to present a clearer discussion of the advantages of their approach.

11.Is it possible to design an experiment with a gold standard where the value of the likelihood can be computed? Then both approaches can be compared against this gold standard.

Reviewer #2: This paper gives a derivation of the marginal likelihood for a general multilevel linear model using a specific prior setting. The derivations look accurate, and related literature is cited accordingly. I have the following comments:

1. Please fix "... the this ..." on line 170.

2. You can skip the equation before Eq. (5).

3. Lines 156 - 159, $\\Sigma_{\\eta}$ is set to $\\nu I$ which implies that the elements of $\\eta$ are independent. What is the implication of this on the multilevel linear model? Is it a restrictive approach? Is it possible to do the derivations if the elements of $\\eta$ are not assumed to be independent?

4. Please link the result in Eq. (7) back to the elements given between lines 120 - 128. Or it would be good to provide an algorithm to outline the implementation of the derivation there in practice. This is only a part of the whole implementation and should be understandably by the practitioners.

5. The example lacks an informative data definition. Some descriptive statistics would help.

6. Please give the data definition and the model's settings starting on line 203 in separate paragraphs for clarity. It is hard to follow in its current form.

7. Once we get Table 1, what will we do with it? The link between the application and model selection is missing. What is the implication of your model selection effort? What does Minnesota radon contamination look like from the perspective of the best model selected by your approach?

8. What if I use an RJMCMC approach? Would I get different results in terms of the estimates of radon contamination? In that sense, what is the contribution of the work to the existing literature?

6. PLOS authors have the option to publish the peer review history of their article (what does this mean?). If published, this will include your full peer review and any attached files.

Reviewer #1: No

Reviewer #2: No

---

## [Author Response · Author response to Decision Letter 0]

14 Nov 2022

Reviewer #1:

1. I’ve thought about this and I’m wary about getting the language right, as I believe ‘marginal partial likelihood’ and ‘partial marginal likelihoods’ are both different things in themselves. We decided to rename this to ‘integrated likelihood’, which has been used to denote this in previous works.

2. Corrected.

3. Corrected.

4. Corrected.

5. No, this is correct as it is currently, though perhaps it could be made clearer, which I have done.

6. Corrected.

7. Corrected.

8. Same as point 5, this is correct as it was.

9. This is a good suggestion, which we should have included from the start and have now implemented.

10. We did consider computation time but decided not to include it in the original manuscript. For the most part, it does offer a moderate improvement, but is of a similar order in magnitude. For the more complicated models where the derived likelihood is sufficiently complicated that sampling on the reduced parameter set takes longer than sampling all parameters. The advantage of this approach is primarily that we believe it is a better and more robust estimate of the model evidence than using all parameters, due to the difficulty in successfully sampling high-dimensional spaces. The simulation study confirms this.

11. This is an interesting idea. I think the answer is no, if the true model is of the same form as those originally described. However, it is possible for a linear model with normal-inverse-gamma joint prior. To allow this comparison, we added this model in the simulation study, however we have also highlighted that the normal-inverse-gamma model is not generally a good model to choose. There are other approaches to computing the model evidence given the marginal likelihood we proposed, but no obvious rationale for choosing that over SMC.

Reviewer #2:

1. Corrected.

2. Not clear which part of the equation array you are referring to, but have removed the first line.

3. This is an example of the form that the covariance could take, which we have not assumed in general. The point here is that since the covariance matrix is positive definite n_j \\times n_j matrix, we can represent it as a vector of length < n_j * (n_j + 1) / 2, which is preferable for computational reasons. I have hopefully made this clearer.

4. The accompanying code is made open-access and available in order to allow implementation by other practitioners. I added a paragraph at the end of Methods to describe how the log marginal likelihood can be used in conjunction with any MCMC sampling scheme. I have also added an extra line after each showing the logarithm of the derived likelihoods, which is used in practice for computational reasons (e.g. underflow). I’m unclear what you are asking in terms of linking (7) back to lines 120-128, but I think the question is how to get from the (integrated) likelihood to the model evidence. This step is not part of our proposal, but I have added references relating to this.

5. We have added some descriptive statistics.

6. Rewritten for greater clarity.

7. We have added a figure showing the Minnesota radon contamination dataset using some of the models, following the same style as https://docs.pymc.io/en/v3/pymc-examples/examples/case_studies/multilevel_modeling.html.

8. The idea of the manuscript is that by first marginalising out a subset of parameters, any subsequent method (particularly MCMC methods) used to estimate the model evidence will result in a more robust estimate, as the sampling is over a hugely reduced dimension. This includes SMC, RJMCMC, hierarchical MCMC i.e. Chib and Carlin. We are agnostic to the choice of method overlayed on top of this, choosing SMC for ease, but we anticipate similar results with other MCMC methods. We believe testing across a wider range of MCMC methods is beyond the scope of what this manuscript seeks to achieve, but the code can easily be adapted to any MCMC sampling method within the PyMC package (I don’t believe this includes RJMCMC currently). The discussion has been amended to make this point clearer.

Thank you to both reviewers for your considered and detailed feedback.

Note to editor:

We have made some other minor changes, including updating and rerunning the code to use PyMC v4 instead of PyMC v3. This has resulted in some changes to Table 1 as, even when seeded, PyMC gives slightly different results when run multiple times.

---

## [Decision Letter · Decision Letter 1]

1 Dec 2022

PONE-D-22-21044R1Bayesian model selection for multilevel models using integrated likelihoodsPLOS ONE

Dear Dr. Edinburgh,

Thank you for submitting your manuscript to PLOS ONE. After careful consideration, we feel that it has merit but does not fully meet PLOS ONE’s publication criteria as it currently stands. Therefore, we invite you to submit a revised version of the manuscript that addresses the points raised during the review process.

We look forward to receiving your revised manuscript.

Kind regards,

Alessandro Barbiero, Ph.D. in Statistics

Academic Editor

PLOS ONE

Journal Requirements:

Reviewers' comments:

Reviewer's Responses to Questions

**Comments to the Author**

1. If the authors have adequately addressed your comments raised in a previous round of review and you feel that this manuscript is now acceptable for publication, you may indicate that here to bypass the “Comments to the Author” section, enter your conflict of interest statement in the “Confidential to Editor” section, and submit your "Accept" recommendation.

Reviewer #1: (No Response)

Reviewer #2: All comments have been addressed

2. Is the manuscript technically sound, and do the data support the conclusions?

Reviewer #1: Yes

Reviewer #2: Yes

3. Has the statistical analysis been performed appropriately and rigorously? 

Reviewer #1: Yes

Reviewer #2: Yes

4. Have the authors made all data underlying the findings in their manuscript fully available?

Reviewer #1: Yes

Reviewer #2: Yes

5. Is the manuscript presented in an intelligible fashion and written in standard English?

Reviewer #1: Yes

Reviewer #2: Yes

6. Review Comments to the Author

Reviewer #1: I thank the authors for addressing my earlier comments. I only one more comment. I like the newly added simulation study as it clearly shows the advantages of the integrated and full likelihood in terms of computation time, accuracy of model evidence and posterior estimates. However, I think the discussion on Table 1 is too brief. There is a paragraph under the section "Discussion" on page 21, lines 410-418 concerning the results on Table 1, and I suggest moving that to the section on "Example: Simulation study".

Reviewer #2: (No Response)

7. PLOS authors have the option to publish the peer review history of their article (what does this mean?). If published, this will include your full peer review and any attached files.

Reviewer #1: No

Reviewer #2: No

---

## [Author Response · Author response to Decision Letter 1]

15 Dec 2022

*New response (15th Dec)

Thank you again to the reviewers and the editor. I have added a few extra sentences to the text in Example: Simulation Study, as requested. I wasn't completely sure what lines from the Discussion you were referring to, as they did not seem to match up with my version of the manuscript, but I think it was clear what you were asking.

I have also checked and amended references, to include the date accessed for websites and having updated the Zenodo repository containing accompanying code.

---

## [Editor Report · Decision Letter 2]

20 Dec 2022

Bayesian model selection for multilevel models using integrated likelihoods

PONE-D-22-21044R2

Dear Dr. Edinburgh,

We’re pleased to inform you that your manuscript has been judged scientifically suitable for publication and will be formally accepted for publication once it meets all outstanding technical requirements.

Kind regards,

Alessandro Barbiero, Ph.D. in Statistics

Academic Editor

PLOS ONE
---

## [Editor Report · Acceptance letter]

2 Feb 2023

PONE-D-22-21044R2 

Bayesian model selection for multilevel models using integrated likelihoods 

Dear Dr. Edinburgh:

I'm pleased to inform you that your manuscript has been deemed suitable for publication in PLOS ONE. Congratulations! Your manuscript is now with our production department. 

Kind regards, 

on behalf of

Dr. Alessandro Barbiero 

Academic Editor

PLOS ONE